# A Tensor-Based Multilayer Graph Representation Learning

### Maolin Wang
City University of Hong Kong
Hong Kong SAR, China
morin.wang@cityu.edu.hk

### Ziting Mai
City University of Hong Kong
Hong Kong SAR, China

### Xuhui Chen
City University of Hong Kong
Hong Kong SAR, China

### Zhiqi Li
City University of Hong Kong
Hong Kong SAR, China

### Tianshuo Wei
City University of Hong Kong
Hong Kong SAR, China

### Yutian Xiao
Beihang University
Beijing, China

### Wenlin Zhang
City University of Hong Kong
Hong Kong SAR, China

### Wanyu Wang
City University of Hong Kong
Hong Kong SAR, China

### Ruocheng Guo
Independent Researcher

### Haoxuan Li
Peking University
Beijing, China

### Zenglin Xu
Fudan University
Shanghai, China

### Xiangyu Zhao
City University of Hong Kong
Hong Kong SAR, China

## Abstract

While traditional network analysis focuses on single-layer networks, real-world systems often form multilayer networks with multiple relationship types. However, existing methods typically fail to capture complex inter-layer dependencies by treating layers independently or aggregating them. To address this, we propose T-GINEE (Tensor-Based Generalized Multilayer-graph Estimating Equation), a statistical regularization framework combining tensor-based generalized estimating equations with task-specific loss to model cross-network correlations explicitly. Key innovations include: (1) CP tensor decomposition capturing structural dependencies via shared latent factors; (2) a generalized estimating equation framework modeling inter-layer correlations through working covariance matrices; and (3) a flexible link function accommodating characteristics like sparsity. Extensive experiments on synthetic and real-world datasets validate T-GINEE's effectiveness for multilayer network analysis.

**ACM Reference Format:**
Maolin Wang, Ziting Mai, Xuhui Chen, Zhiqi Li, Tianshuo Wei, Yutian Xiao, Wenlin Zhang, Wanyu Wang, Ruocheng Guo, Haoxuan Li, Zenglin Xu, and Xiangyu Zhao. 2026. A Tensor-Based Multilayer Graph Representation Learning . In *Proceedings of The 1st Workshop on Interplay Between Classical Tensor Methods and Foundation Models (TensorKDD '26).* ACM, New York, NY, USA, 5 pages. https://doi.org/10.1145/nnnnnnn.nnnnnnn

## 1 Introduction

In the real world, interactions between entities are often multi-faceted, with these multi-relational characteristics engaging one another under varied circumstances or through distinct modalities. For instance, in social networks [26], individuals may be connected through multiple relationship types such as friends, colleagues, and family. In biology [32], genes or proteins exhibit various collaboration schemes like co-expression and physical interactions. In global trade, countries exchange a wide range of different commodities.

For such intricate relational landscapes, a multi-layer graph offers a faithful and structured representation. This architecture is defined by a common set of vertices, where each layer is endowed with a unique edge set to delineate a specific type of relation. Such graphs are prevalent across numerous disciplines, including social graphs that capture multiple interaction channels between individuals [8], biological graphs detailing different collaboration schemes among genes or proteins [17, 19], and global trade graphs mapping the exchange of various commodities [2, 22]. To effectively analyze these intricate structures, a fundamental step is to learn low-dimensional vector representations (i.e., embeddings) for the entities that capture the complex relational information encoded across layers.

Numerous approaches have been developed for graph embedding, employing various techniques such as similarity indices [3], maximum likelihood models [30], matrix factorization [5, 7, 25], and graph neural networks [9, 13, 28]. For multilayer graph embedding, which provides a richer representation of complex systems [14], analysis often involves extending these single-layer techniques. Prominent approaches include tensor-based methods that leverage the natural tensor structure of multilayer graphs [1, 15], as well as adaptations of deep learning models like GCNs and random-walk embeddings [6, 24].

However, a critical challenge underlying many of these methods is the lack of a rigorous theoretical foundation for the multilayer context. While embedding learning has proven effective for single-layer graphs [4], we lack robust theoretical frameworks systematically characterizing the embedding process across multiple layers [11]. This absence of formal tools describing how embeddings capture and preserve cross-layer dynamics significantly impedes developing principled approaches, representing a fundamental limitation in the field [12, 20, 23].

Without this theoretical guidance, existing approaches often resort to simplistic solutions, such as learning representations for layers independently [5, 25] or using basic aggregation techniques [16, 21]. These methods lack the grounding to explain how embeddings should optimally encode the nuanced ways in which relationships in one layer might influence or contradict another [18, 31]. This deficit is especially problematic for real-world systems where entities engage through multiple relation types simultaneously [27, 29], highlighting the urgent need for new frameworks that can faithfully represent this complex interplay [10, 23].

To address these challenges, we propose T-GINEE (Tensor-based Generalized Multilayer-graph Estimating Equation), a statistical regularization framework that combines tensor-based generalized estimating equations with task-specific loss to explicitly model cross-network correlations. The key technical innovations of T-GINEE include: (1) A CP tensor decomposition approach that effectively captures structural dependencies through shared latent factors while maintaining computational efficiency; (2) A generalized estimating equation framework that explicitly models the correlations between different network layers through working covariance matrices; and (3) A flexible link function design that accommodates various network characteristics, including sparsity. Unlike previous approaches that rely on simple aggregation or separate modeling [16, 21, 25], T-GINEE provides a principled statistical framework to jointly model multiple networks.

Overall, T-GINEE integrates a symmetric CP tensor decomposition with a generalized estimating equation (GEE) formulation, provides asymptotic statistical guarantees, and validates the framework on both synthetic and real-world multilayer networks.

## 2 Methodology

In this section, we present Tensor-based Generalized Estimating Equations (T-GINEE), a framework for learning embeddings from multilayer graphs. Our method combines a low-rank CP parameterization of a multilayer graph with a generalized estimating equations (GEE) estimator, equipped with a structured working covariance, to jointly model within-layer and cross-layer dependencies.

### 2.1 Overview

Real-world networks often exhibit complex interdependencies, where multiple network structures coexist and influence each other. For instance, an individual's friendship networks on multiple social media platforms (such as Facebook, LinkedIn, and TikTok) form correlated multilayer graphs over the same set of users. We propose a statistical regularization framework that leverages tensor-based generalized estimating equations to explicitly model such cross-network correlations. Our framework, referred to as T-GINEE, consists of several core components: (i) a symmetric CP decomposition of the parameter tensor $\Theta$ of the multilayer graph into node embeddings $\alpha$ and layer-specific embeddings $\beta$; (ii) the construction of a parameter vector $\gamma$ from these embeddings; and (iii) a tensor-based GEE formulation that estimates $\gamma$ under a working covariance model and thereby captures complex dependencies across layers.

## 2.2 Problem formulation

Consider a multilayer network/graph $\mathcal{G} = (\mathcal{V}, \{\mathcal{G}^{(m)}\}_{m=1}^{M})$, where $\mathcal{V} = \{v_1, \ldots, v_n\}$ is a common set of vertices that interact across $M$ different but potentially correlated network layers. Each layer $\mathcal{G}^{(m)} = (\mathcal{V}, \mathcal{E}^{(m)})$ captures a distinct type of relationship. We focus on undirected networks and index undirected edges by unordered pairs $\{i, j\}$, using the convention that $i \leq j$. In our notation we include pairs with $i = j$; in the empirical datasets we consider, the diagonal entries are identically zero (i.e., $\mathcal{A}_{i,i,m} \equiv 0$), so allowing $i = j$ does not affect estimation but simplifies asymptotic analysis.

Let $\mathcal{A} \in \{0, 1\}^{n \times n \times M}$ be the adjacency tensor, where $\mathcal{A}_{i,j,m} = 1$ indicates an edge of type $m$ between nodes $i$ and $j$, and $\mathcal{A}_{i,j,m} = 0$ otherwise. For each pair $(i, j)$ with $i \leq j$, the vector $\mathcal{A}_{i,j,\cdot} \in \mathbb{R}^M$ is treated as an $M$-dimensional binary response with mean $\mathcal{P}_{i,j,\cdot}(\Theta)$ and covariance matrix $\Sigma_{i,j}$. In line with the GEE paradigm, we specify only its first two moments, that is $\mathbb{E}[\mathcal{A}_{i,j,\cdot}] = \mathcal{P}_{i,j,\cdot}(\Theta)$ and $\mathrm{Cov}(\mathcal{A}_{i,j,\cdot}) = \Sigma_{i,j}$, without imposing a full joint distribution. Conditional on the parameters, we assume that the edge vectors $\{\mathcal{A}_{i,j,\cdot} : i \leq j\}$ are independent across different node pairs $(i, j)$. Within each pair $(i, j)$, the components $\mathcal{A}_{i,j,1}, \ldots, \mathcal{A}_{i,j,M}$ may be correlated and this dependence is captured by $\Sigma_{i,j}$.

The parameter tensor $\Theta \in \mathbb{R}^{n \times n \times M}$ is linked to the mean tensor $\mathcal{P}$ through a known three-times continuously differentiable link function $g$, applied elementwise, such that

$$\mathcal{P}_{i,j,\cdot} = g^{-1}(\Theta_{i,j,\cdot}).$$

Typical choices of the link function include:

- the identity link $g(x) = x$, primarily used for weighted networks where $\mathcal{A}_{i,j,m}$ need not be binary; for Bernoulli observations, the identity link can be applied by implicitly constraining $\Theta_{i,j,m} \in [0, 1]$;
- the probit link $g(x) = \Phi^{-1}(x)$ with inverse $g^{-1}(x) = \Phi(x) = \int_{-\infty}^{x} (2\pi)^{-1/2} \exp(-t^2/2) \, \mathrm{d}t$;
- the logit link $g(x) = \log\{x/(1-x)\}$ with inverse $g^{-1}(x) = 1/(1 + e^{-x})$;
- a sparsity-aware logit $g(x) = \log\{x/(s-x)\}$ with inverse $g^{-1}(x) = \frac{s}{1 + e^{-x}}$, where $0 < s < 1$ is a sparsity coefficient that can decrease with $n$ and $M$ to accommodate increasingly sparse networks. When $s$ is allowed to depend on $(n, M)$, we assume there exists $\varepsilon > 0$ such that, for all $(i, j, m)$ and all $\gamma$ in a neighborhood of $\gamma_0$,

$$\varepsilon \leq \mathcal{P}_{i,j,m}(\gamma) \leq s - \varepsilon,$$

which in particular guarantees that $g'$ and the required higher-order derivatives remain uniformly bounded.

### 2.3 Low-rank tensor decomposition

Before introducing the formal CP decomposition, we briefly motivate this modeling choice. In many multilayer systems, the same set of nodes participates in different relation types (layers) through a shared, low-dimensional set of latent traits (e.g., social roles, functional modules, or economic profiles). A symmetric CP factorization of the parameter tensor $\Theta$ reflects this idea: the node embeddings $\alpha$ define a common latent space across all layers, while the layer embeddings $\beta$ determine how each relation type weights these latent

factors. This yields a compact representation that captures higher-order interactions across nodes and layers, reduces the number of free parameters, and aligns with empirical findings that a relatively small number of latent dimensions often suffices to explain multilayer network structure.

In our asymptotic analysis, we consider a regime where $n$ grows while $M$ and $R$ are either fixed or increase sufficiently slowly, more precisely satisfying $(n + M)R = o(n^{1/3})$. To effectively model structural dependencies while maintaining computational efficiency, we employ a symmetric CP decomposition for the parameter tensor $\Theta$:

$$\Theta = \mathcal{I} \times_1 \alpha \times_2 \alpha \times_3 \beta = \sum_{r=1}^{R} \alpha^{(r)} \circ \alpha^{(r)} \circ \beta^{(r)}, \quad (1)$$

where $\mathcal{I} \in \mathbb{R}^{R \times R \times R}$ denotes the order-3 identity tensor used in CP decomposition, $\alpha \in \mathbb{R}^{n \times R}$ contains node embeddings and $\beta \in \mathbb{R}^{M \times R}$ contains layer-specific embeddings. Here $\alpha^{(r)}$ and $\beta^{(r)}$ denote the $r$-th columns of $\alpha$ and $\beta$, respectively, and $\circ$ denotes the outer product. This parameterization enforces $\Theta_{i,j,m} = \Theta_{j,i,m}$ for all $i, j, m$, consistent with undirected layers, and inherently imposes structural constraints through shared latent factors.

For optimization purposes, we vectorize these factor matrices into a compact representation:

$$\gamma = \left[ \text{vec}(\alpha)^\top, \ \text{vec}(\beta)^\top \right]^\top \in \mathbb{R}^{(n+M)R}. \quad (2)$$

This parameter vector $\gamma$ encapsulates the essential cross-layer dependencies and serves as the decision variable in our estimating equations. The low-rank tensor decomposition offers several advantages: it significantly reduces the number of free parameters, improving computational efficiency and mitigating overfitting; by sharing latent factors across dimensions, it naturally captures the inherent relationships between nodes and layers; and it yields interpretable components, where $\alpha$ represents node-level patterns and $\beta$ captures layer-level characteristics.

## 2.4 Tensor-based statistical regularization

We now introduce the tensor-based generalized estimating equations that define T-GINEE. The tensor-based estimating equations for multilayer graphs are

$$\sum_{i \leq j} \left( \frac{\partial \mathcal{P}_{i,j,\cdot}}{\partial \gamma} \right)^\top \widehat{\Sigma}_{i,j}^{-1} \left( \mathcal{A}_{i,j,\cdot} - \mathcal{P}_{i,j,\cdot}(\gamma) \right) = \mathbf{0}, \quad (3)$$

where $\widehat{\Sigma}_{i,j}$ is a working covariance matrix. We denote the left-hand side of (3) by $s(\gamma)$; thus T-GINEE seeks a root $\hat{\gamma}$ of $s(\gamma) = \mathbf{0}$.

To compute the score contributions, we first derive $\partial \mathcal{P}_{i,j,\cdot} / \partial \gamma$ via the chain rule: starting from the Jacobian $\partial \text{vec}(\Theta) / \partial \gamma$ implied by the CP decomposition, then applying the derivative of the link function, and finally projecting onto the $(i, j)$-th fibers using basis tensors $\mathcal{E}^{(i,j,m)}$.

Specifically, since $\mathcal{P}_{i,j,m} = g^{-1}(\Theta_{i,j,m})$ and $g$ is differentiable, we have

$$\frac{\partial \mathcal{P}_{i,j,\cdot}}{\partial \gamma} = \left[ \text{diag}\left( g'(\mathcal{P}_{j,1}), \ldots, g'(\mathcal{P}_{j,M}) \right) \right]^{-1} \frac{\partial \Theta_{i,j,\cdot}}{\partial \gamma} \quad (4)$$

where $g'$ denotes the derivative of $g$, applied elementwise, and the diagonal matrix has $(m, m)$-entry $g'(\mathcal{P}_{i,j,m})$ for $m \in [M]$. Let

$\mathcal{E}^{(i,j,m)} \in \mathbb{R}^{n \times n \times M}$ be the tensor unit whose $(i', j', m')$-th entry is $\mathbf{1}\{(i, j, m) = (i', j', m')\}$. Then

$$\frac{\partial \Theta_{i,j,\cdot}}{\partial \gamma} = \left[ \text{vec}(\mathcal{E}^{(i,j,1)}), \ldots, \text{vec}(\mathcal{E}^{(i,j,M)}) \right]^\top \frac{\partial \text{vec}(\Theta)}{\partial \gamma} \quad (5)$$

Under the CP decomposition of $\Theta$, the Jacobian matrix with respect to the parameter vector $\gamma$ takes the block form

$$\frac{\partial \text{vec}(\Theta)}{\partial \gamma} = \begin{bmatrix} (\beta^{(1)})^\top \otimes \left( I_n \otimes \alpha^{(1)} + \alpha^{(1)} \otimes I_n \right) \\ \vdots \\ (\beta^{(R)})^\top \otimes \left( I_n \otimes \alpha^{(R)} + \alpha^{(R)} \otimes I_n \right) \\ I_M \otimes \left( \alpha \odot_{\text{KR}} \alpha \right) \end{bmatrix}, \quad (6)$$

where $\otimes$ denotes the Kronecker product, and $\odot_{\text{KR}}$ denotes the Khatri–Rao (column-wise Kronecker) product:

$$\alpha \odot_{\text{KR}} \alpha = \left[ \alpha^{(1)} \otimes \alpha^{(1)}, \ldots, \alpha^{(R)} \otimes \alpha^{(R)} \right] \in \mathbb{R}^{n^2 \times R}.$$

Each of the first $R$ block rows in (6) has dimension $(n^2 M) \times n$, and the last block row $I_M \otimes \left( \alpha \odot_{\text{KR}} \alpha \right)$ has dimension $(n^2 M) \times (MR)$, so that $\partial \text{vec}(\Theta) / \partial \gamma \in \mathbb{R}^{(n^2 M) \times (n+M)R}$.

## 2.5 Covariance structure and estimation

The cross-layer dependencies within each node pair $(i, j)$ are summarized by the covariance matrices $\Sigma_{i,j} = \text{Cov}(\mathcal{A}_{i,j,\cdot})$. In line with the GEE framework, we approximate these true covariances by a parsimonious working covariance family that assumes a common correlation structure across node pairs:

$$\Sigma_{i,j}^{\text{w}}(\gamma) = \Gamma_{i,j}^{1/2} W \Gamma_{i,j}^{1/2}, \quad (7)$$

where $\Gamma_{i,j} \in \mathbb{R}^{M \times M}$ is a diagonal matrix whose $(m, m)$-th entry is $\mathcal{P}_{i,j,m}(\gamma)\left(1 - \mathcal{P}_{i,j,m}(\gamma)\right)$, and $W \in \mathbb{R}^{M \times M}$ is a positive-definite correlation matrix shared across node pairs. The working covariance matrices appearing in (3) are then

$$\widehat{\Sigma}_{i,j} = \Gamma_{i,j}^{1/2} \widehat{W} \Gamma_{i,j}^{1/2},$$

where $\widehat{W}$ is an empirical estimate of $W$.

We estimate $W$ by pooling residuals across all node pairs:

$$\widehat{W} = \frac{1}{N} \sum_{i \leq j} \Gamma_{i,j}^{-1/2} \left( \mathcal{A}_{i,j,\cdot} - \mathcal{P}_{i,j,\cdot}(\hat{\gamma}) \right) \left( \mathcal{A}_{i,j,\cdot} - \mathcal{P}_{i,j,\cdot}(\hat{\gamma}) \right)^\top \Gamma_{i,j}^{-1/2} \quad (8)$$

where $\hat{\gamma}$ is the current estimate of $\gamma$, $N = n(n + 1)/2$ is the number of node pairs with $i \leq j$, and the sum runs over all such pairs, consistent with our convention in Section 2.3. Under mild regularity conditions, the eigenvalues of $\Sigma_{i,j}^{\text{w}}(\gamma)$ are bounded away from zero and infinity, which ensures numerical stability. In practice, to avoid numerical instability when inverting $\widehat{\Sigma}_{i,j}$, we may add a small ridge term $\epsilon I_M$ to $\widehat{W}$.

## 2.6 Optimization and computational complexity

In practice, we do not solve the nonlinear estimating equations (3) in closed form. Instead, we optimize $\gamma$ using iterative gradient-based updates, alternating with periodic updates of the working correlation matrix $W$.

At a high level, each training epoch consists of: (i) computing the CP-based parameter tensor $\Theta$ and the corresponding edge probabilities $\mathcal{P}_{i,j,m}(\gamma)$ for sampled node pairs $(i, j)$ and layers $m$; (ii) evaluating the score contributions in (3) via the Jacobian $\partial \mathcal{P}_{i,j,\cdot} / \partial \gamma$; and (iii) updating $\gamma$ by a gradient step (with optional regularization),

followed by an update of $W$ using (8). In our implementation we initialize $W$ as the identity $I_M$ (an independence working structure) and start updating it after a few warm-up epochs.

In most real-world settings, multilayer networks are sparse. Using sparse tensor representations and mini-batching over observed edges, the dominant cost per iteration scales as $O(R|E|)$, where $|E|$ is the total number of observed edges across all layers. The update of the working correlation $W$ in (8) is computed from aggregated residuals and can be performed infrequently (e.g., every $K$ epochs), so its amortized overhead is small compared to embedding updates.

## 3 Experiments

In this section, we conduct comprehensive experiments to evaluate our T-GINEE framework.

### 3.1 Synthetic Data Results

To evaluate our method in a controlled environment, we generate synthetic multilayer networks with known correlation structures using a parameterized model:

$$\mathcal{P}_{i,j,m} = \rho \cdot \mathcal{P}_{i,j}^{\text{base}} + (1 - \rho) \cdot U_{i,j,m}, \qquad \mathcal{A}_{i,j,m} = \mathbf{1}\{V_{i,j,m} < \mathcal{P}_{i,j,m}\}, \tag{9}$$

Here, $\rho = 0.2$ controls inter-layer correlation, $\mathcal{P}_{i,j}^{\text{base}}$ is a shared base probability matrix, and $U_{i,j,m}$ and $V_{i,j,m}$ are i.i.d. Unif$[0, 1]$ noise variables. We construct networks with $n = 100$ nodes and $M = 3$ layers for link prediction tasks.

As shown in Table 1, T-GINEE achieves the highest AUC score of 0.9395, substantially outperforming all baselines including the second-best HOSVD. The significant performance gap between tensor-based methods and simpler approaches like LSE (0.2234) and MASE (0.3821) confirms the importance of explicitly modeling multilayer dependencies. Furthermore, the dramatic improvement of T-GINEE over basic CP decomposition (0.4488) demonstrates the effectiveness of our statistical regularization framework in capturing complex inter-layer correlations.

### 3.2 Real-World Results

Based on the experimental results shown in Table 2, our proposed T-GINEE model demonstrates superior performance across datasets of varying scales and complexities compared to baseline methods. On the standard benchmark datasets (AUCS, Krackhardt, WAT, and Yeast), T-GINEE achieves the highest AUC scores of 0.920, 0.948, 0.838, and 0.921 respectively, outperforming all baseline methods in our experiments. Among the baseline methods, traditional matrix factorization approaches, such as SVD and NMF, show relatively strong performance, with SVD achieving the second-best results on AUCS (0.877) and Krackhardt (0.932). HOSVD, as a tensor-based method, also demonstrates competitive performance, particularly on the AUCS and Yeast datasets. However, simpler methods such as CP decomposition and LSE exhibit limited effectiveness, with LSE performing poorly on Yeast.

To comprehensively evaluate the scalability of T-GINEE, we extended our experiments to two large-scale real-world networks: DBLP, an academic collaboration network with up to 300,000 nodes, and Stack Overflow, a massive temporal interaction network with approximately 2.6 million nodes. As indicated in the rightmost

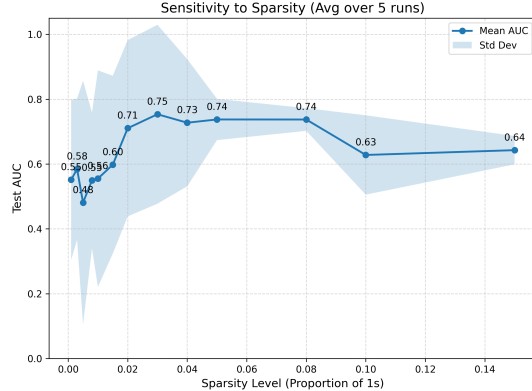

**Figure 1: Sensitivity of T-GINEE to graph sparsity on synthetic multilayer networks. We vary the proportion of observed edges (proportion of $1$ entries).**

columns of Table 2, most traditional tensor-based methods, including CP, Pure-Tucker, HOSVD, and MASE, failed to process these large-scale datasets, resulting in Out-Of-Memory (OOM) errors. This highlights the severe computational bottleneck of standard tensor decompositions when applied to million-node graphs. In contrast, T-GINEE successfully scaled to these massive datasets while maintaining superior accuracy. While matrix-based methods like SVD and NMF could handle the scale, T-GINEE surpassed them in performance. On the DBLP dataset, T-GINEE achieved an AUC of 0.6478, outperforming SVD (0.6093). On the massive Stack Overflow dataset, T-GINEE maintained high accuracy (0.9831), exceeding the best matrix baselines.

### 3.3 Sensitivity to Sparsity

We further study how T-GINEE performs as multilayer networks vary in sparsity. We adjust the Bernoulli generator so the proportion of observed edges (i.e., $1$ entries in the adjacency tensor) ranges from below 1% to about 15%. At each sparsity level, we run 5 trials with different seeds. Figure 1 reports mean test AUC with one standard deviation. AUC increases from the extremely sparse regime to moderate sparsity, peaking at roughly AUC $\approx 0.75$, then remains fairly stable and declines only gradually for dense graphs, staying above AUC $\approx 0.63$ even at $10$–$15$% edge density. Overall, T-GINEE is robust across a broad sparsity range: performance is weaker and more variable when edges are extremely sparse, but stabilizes once a moderate amount of edge information is available.

## 4 Conclusion

We propose T-GINEE, a tensor-based generalized estimating equation framework for multilayer graph representation learning that explicitly models cross-network dependencies through a principled statistical formulation. By combining CP decomposition with GEE, T-GINEE makes a central theoretical contribution: establishing the consistency and asymptotic normality of embeddings under mild regularity conditions, thereby providing rigorous statistical guarantees. Experiments demonstrate its effectiveness on synthetic and

**Table 1: Link prediction performance (AUC) on synthetic multilayer network.**

| Method | CP | Tucker | NMF | SVD | LSE | MASE | NNTUCK | SPECK | HOSVD | T-GINEE |
|---|---|---|---|---|---|---|---|---|---|---|
| AUC | 0.4488 | 0.5291 | 0.7216 | 0.8130 | 0.2234 | 0.3821 | 0.6105 | 0.7603 | 0.8503 | **0.9395** |

**Table 2: Performance comparison of different methods. "oom" denotes Out-Of-Memory errors.**

| Method | AUC score on different datasets | | | | | |
|---|---|---|---|---|---|---|
| | AUCS | Krackhardt | WAT | Yeast | dblp | stackoverflow |
| CP | 0.374 | 0.354 | 0.454 | 0.397 | oom | oom |
| Tucker | 0.487 | 0.702 | 0.580 | 0.745 | oom | oom |
| NMF | 0.848 | 0.921 | 0.707 | 0.863 | 0.6505 | 0.9642 |
| SVD | 0.877 | 0.932 | 0.719 | 0.879 | 0.6093 | 0.9682 |
| LSE | 0.297 | 0.384 | 0.153 | 0.047 | 0.6302 | oom |
| MASE | 0.480 | 0.361 | 0.342 | 0.347 | oom | oom |
| NNTUCK | 0.500 | 0.521 | 0.741 | 0.667 | oom | oom |
| SPECK | 0.793 | 0.658 | 0.655 | 0.903 | oom | oom |
| HOSVD | 0.897 | 0.783 | 0.820 | 0.902 | oom | oom |
| T-GINEE | **0.920** | **0.948** | **0.838** | **0.921** | 0.6478 | **0.9831** |

real-world networks, highlighting the robust mathematical foundation T-GINEE offers for analyzing complex interdependent systems. Limitations include potential constraints on extremely sparse or large-scale networks and a priority on statistical validation over engineering optimizations like attention-based architectures. Future work will explore combining our objective with deep encoders and extending the framework to dynamic settings.

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
