# OpenReview forum: "A Tensor-Based Multilayer Graph Representation Learning"
_KDD.org/2026/Workshop/TensorKDD — KDD 2026 Workshop TensorKDD Oral_

### Official Review · Reviewer_X2hy · 2026-06-09
**Clear Motivation and Interesting Framework, but Limited Experimental Validation**

**Rating:** Accept
**Confidence:** 4
**Best Paper Recommendation:** No

**Review:**

**Paper Summary**

The paper proposes T-GINEE, a tensor-based generalized estimating equation framework for multilayer graph representation learning. Its main goal is to capture cross-layer dependencies that are often ignored by methods that model layers independently. The method combines symmetric CP decomposition with a working covariance matrix to model node-layer interactions and inter-layer correlations. Experiments on synthetic and real-world link prediction benchmarks show promising AUC performance against classical matrix- and tensor-based baselines.

**Paper Strengths**

**S1.** The motivation is clear. Multilayer graphs naturally contain dependencies across different relation types, and treating layers independently or simply aggregating them can lose important structural information. The paper clearly identifies this limitation and positions cross-layer dependency modeling as the central objective.

**S2.** The proposed combination of CP decomposition and generalized estimating equations is reasonably well motivated. The CP decomposition provides a compact low-rank representation of node-layer interactions, while the working covariance matrix offers a principled way to model correlations between layers.

**S3.** The empirical results are promising. T-GINEE achieves the best or highly competitive AUC scores across several synthetic and real-world datasets.

**Paper Weaknesses**

**W1.** The experimental evaluation is somewhat limited and may be considered outdated. The baselines are mostly classical matrix and tensor factorization methods, such as CP, Tucker, NMF, SVD, and HOSVD. However, stronger modern graph representation learning baselines, such as relational GNNs, graph transformers, graph autoencoders, or neural link prediction models, are missing. This weakens the claim that T-GINEE achieves state-of-the-art performance in multilayer graph representation learning. Also, the paper should clarify how the train/validation/test split was performed.

**W2.** The theoretical contributions are not sufficiently substantiated in the current manuscript. The paper claims consistency and asymptotic normality, but formal theorem statements, assumptions, proof sketches, and clarification of identifiability issues are largely missing. Since CP decomposition has scale, sign, and permutation ambiguities, the paper should clarify how these issues are handled in the theoretical analysis.

**Summary**

I lean toward acceptance for a workshop. The paper addresses a clear and meaningful problem, and the proposed CP + GEE formulation is interesting. Although the experimental comparison is somewhat limited and the current baselines are not fully up to date, the paper’s objective is well defined and the statistical modeling perspective provides a useful contribution. I would encourage the authors to strengthen the experimental section, include modern graph representation learning baselines, provide ablation studies for the covariance component, and clarify the theoretical claims in more detail.

---

### Official Review · Reviewer_ioKP · 2026-06-10

**Rating:** Accept
**Confidence:** 4
**Best Paper Recommendation:** Yes

**Review:**

This paper proposes a tensor-based framework for multilayer graph representation learning called T-GINEE that combines symmetric CP decomposition with generalized estimating equations (GEE). T-GINEE represents a multilayer graph as an adjacency tensor and models edge probabilities through shared node embeddings and layer-specific factors, while using a working covariance matrix to capture cross-layer dependencies within each node pair. Experiments were performed on synthetic and real-world datasets and compared against tensor methods, matrix factorization, and spectral baselines. The reported results suggest that T-GINEE can improve AUC on several benchmark datasets and scale to large networks

$\textbf{Strengths:}$
- The paper addresses an important problem, which is learning representations for multilayer graphs while modeling dependencies across layers.
- The combination of CP with a GEE-style covariance structure is interesting and gives the paper a clearer statistical motivation.
- T-GINEE scales well on larger graphs in contrast to the baselines, which run out-of-memory.

$\textbf{Weaknesses:}$
- Some details are missing for the experimental section, like the training hyperparameters (e.g., learning rate, batch size and so on), dataset statistics, if the sparse formulation of CP was applied (if so, was negative sampling employed), training epochs, and stopping criteria.
-  The paper claims theoretical guarantees, including consistency and asymptotic normality, but no theorem, proof sketch, assumptions, or clear statement of the result is included in the main text
-  The relationship between the GEE estimating equation and the actual training objective is unclear. The paper says it combines tensor-based GEE with task-specific loss, but the task-specific loss isn’t clear.

$\textbf{Things to improve:}$
- The paper should define the learning objective. Is the model solving the estimating equation directly, minimizing the norm of the GEE score, optimizing a likelihood-like loss, or adding the GEE term as a regularizer to a link prediction loss?
- It is unclear where the o(n^1/3) comes from in the asymptotic condition.
- It would be good to explain how link prediction is evaluated. It should say if edges are randomly split, temporally split, or layer-wise split, and whether the test set contains sampled non-edges.
- The conclusion says the method establishes consistency and asymptotic normality, but the paper does not show these results.

---

### Official Review · Reviewer_2kcs · 2026-06-11
**Tensor-GEE formulation with some rudimentary empirical validation**

**Rating:** Accept
**Confidence:** 3
**Best Paper Recommendation:** No

**Review:**

This paper proposes T-GINEE, a tensor-based generalized estimating equation (GEE) framework for multilayer graph representation learning which combines a low-rank CP parameterization of a multilayer graph with a generalized estimating equations
estimator. The method represents a multilayer graph as an adjacency tensor, parameterizes the tensor using a symmetric CP decomposition and uses a GEE-style working covariance matrix to model correlations across layers. The paper evaluates the method mainly on link prediction tasks over synthetic and real-world multilayer networks.

The problem is important, and the proposed framing is potentially valuable. Explicitly modeling cross-layer dependence is valuable as combination of CP factorization with a GEE-style covariance model is a reasonable choice.

The baseline set is not strong enough to corroborate the paper’s claims. The paper mainly compares mainly against classical  factorization methods such as CP, Tucker, NMF, SVD, LSE, MASE, NNTUCK, SPECK, and HOSVD etc. Recent multilayer or multiplex neural link prediction methods should be included or at least discussed, especially methods such as "Link Prediction on Multilayer Networks through Learning of Within-Layer and Across-Layer Node-Pair Structural Features and
Node Embedding Similarity", "Link Prediction in Multilayer Networks via Cross-Network Embedding", MultiSAGE, etc. The paper talks about the literature at a high level but doesn't pick them as a baseline. The paper asserts some theoretical limitations of these methods but doesn't empirically verify the necessity/correctness/value of such a theoretical claim.